# The Relationship between Plasma Alpha-1-Antitrypsin Polymers and Lung or Liver Function in ZZ Alpha-1-Antitrypsin-Deficient Patients

**DOI:** 10.3390/biom12030380

**Published:** 2022-02-28

**Authors:** Annelot D. Sark, Malin Fromme, Beata Olejnicka, Tobias Welte, Pavel Strnad, Sabina Janciauskiene, Jan Stolk

**Affiliations:** 1Department of Pulmonology, Leiden University Medical Center, Member of European Reference Network Lung, Section Alpha-1-Antitrypsin Deficiency, 2333 ZA Leiden, The Netherlands; a.d.sark@lumc.nl; 2Medical Clinic III, Gastroenterology, Metabolic Diseases and Intensive Care, University Hospital RWTH Aachen, 52074 Aachen, Germany; mfromme@ukaachen.de (M.F.); pstrnad@ukaachen.de (P.S.); 3Department of Respiratory Medicine, Biomedical Research in Endstage and Obstructive Lung Disease Hannover (BREATH), Member of the German Center for Lung Research (DZL), 30625 Hannover, Germany; beata.olejnicka@med.lu.se (B.O.); welte.tobias@mh-hannover.de (T.W.); janciauskiene.sabina@mh-hannover.de (S.J.)

**Keywords:** alpha1-antitrypsin, SERPINA1, deficiency, polymers, ELISA, lung function, liver biomarkers

## Abstract

Alpha-1-Antitrypsin (AAT) is a protein of the SERPINA1 gene. A single amino acid mutation (Lys342Glu) results in an expression of misfolded Z-AAT protein, which has a high propensity to intra- and extra-cellular polymerization. Here, we asked whether levels of circulating Z-AAT polymers are associated with the severity of lung disease, liver disease, or both. We obtained cross sectional data from the Dutch part of the Alpha1 International Registry of 52 ZZ-AAT patients who performed a pulmonary function test and donated a blood sample on the same day. From the Alpha-1 Liver Aachen Registry, we obtained a cohort of 40 ZZ-AAT patients with available data on their liver function. The levels of plasma Z-AAT polymers were determined using a LG96 monoclonal antibody-based sandwich ELISA. In a Dutch cohort, the median plasma level of Z-AAT polymers of patients diagnosed for pulmonary disease was 947.5 µg/mL (733.6–1218 µg/mL (95% CI)), which did not correlate with airflow obstruction or gas transfer value. In the Alpha-1 liver patient cohort, the median polymer level was 1245.9 µg/mL (753–2034 µg/mL (95% CI)), which correlated with plasma gamma-glutamyl transferase (GGT, rs = 0.57, *p =* 0.001), glutamate dehydrogenase (GLDH, rs = 0.48, *p* = 0.002) and triglycerides (TG, rs = 0.48, *p* = 0.0046). A Wilcoxon rank test showed higher Z-AAT polymer values for the liver over the lung group (*p* < 0.0001). These correlations support a possible link between plasma Z-AAT polymers and the liver function.

## 1. Introduction

Human alpha-1-antitrypsin (AAT), a member of the serpin (serine protease inhibitor) superfamily, is an acute phase glycoprotein and an inhibitor of neutrophil serine proteases, like elastase, proteinase 3, and cathepsin G, and non-serine proteases, such as caspases and metalloproteases [1]. Like other positive acute phase proteins, increased levels of AAT during inflammation intend to prevent the development of chronic inflammation and organ injury. In vitro and in vivo data provide evidence that AAT expresses broad anti-inflammatory properties, some of which are not related to its anti-protease role [1,2].

AAT is encoded by the SERPINA1 gene and is mainly produced and released into the bloodstream by hepatocytes. Over 200 mutations have been identified in the SERPINA1 gene, which may affect the concentration, functionality, or both of AAT protein [3]. The Z (Glu342Lys, also known as Glu366Lys) and S (Glu264Val, also known as Glu288Val) mutations are the most common and clinically significant, and both are known to cause AAT deficiency (AATD) in the circulation [4].

Individuals homozygous for the Z allele (about 1:2000 to 1:5000 in populations of Northwest European descent) have circulating levels of monomeric AAT that are reduced by about 90% (normal is 1–2 g/L). The ZZ-AATD occurs not because of the lack of AAT protein biosynthesis, but because of Z-AAT intracellular and extracellular polymerization and a diminished secretion [5,6]. It is suggested that a mechanistic link exists between intracellular accumulation of misfolded Z-AAT protein (gain-of-function) and lower secretion of misfolded Z-AAT protein (loss-of-function), leading to tissue damage and eventually development of chronic disease [4,7].

Since the liver produces about 90% of AAT in our body, intrahepatic Z-AAT retention can cause liver damage with variable clinical presentations ranging from neonatal cholestasis to liver cirrhosis and hepatocellular carcinoma in adults [8]. On the other hand, the decreased protease inhibitory capacity normally provided by AAT shifts the protease/anti-protease balance towards proteolysis and increases the risk of developing early onset emphysema, especially in cigarette smokers with severe AATD [9]. Additionally, plasma from all AATD individuals carrying the Z allele of AAT contains significant amounts of circulating AAT polymers [10,11]. These polymers have also been detected in the lung, kidney, and skin [12,13]. While the pathogenic importance of Z-AAT polymers remains to be fully clarified, previous studies have shown that Z-polymers of AAT are present in the bronchial lavage of ZZ AATD individuals and act as chemoattractant for neutrophils in vitro [6]. Other studies have demonstrated that neutrophils and polymeric Z-AAT co-localize in the alveolar wall [13]. A direct relationship between Z-AAT polymers and the excess of neutrophils in the lungs has been suggested [14]. The latter, together with impaired anti-protease activity of AAT, may enhance susceptibility to the development of emphysema [15]. Studies using anti-Z-polymer-specific 2C1 monoclonal antibody have shown that there are multiple forms of polymers [16,17].

Because of the above-described data, circulating polymers constitute attractive biomarkers of the AATD-associated diseases. Indeed, exploratory data suggested a possible association between the levels of circulating Z polymers and lung/liver disease [11]. Current data suggest that polymer levels can reflect lung function decline in AATD patients [18]. Nevertheless, further studies are warranted to establish the value of circulating polymers as prognostic biomarkers for lung and liver diseases. In this study, we assessed whether circulating plasma Z-AAT polymers in patients with ZZ-AATD are associated with pulmonary or liver function.

## 2. Materials and Methods

### 2.1. Patients

From 1999 onwards, informed consent was obtained from patients who had visited our Outpatient Clinic with a stable clinical condition in the past 4 weeks for registration in the Alpha-1 International Registry [19]. Participants have signed an informed consent for unlimited sample analyses, which are related to AATD and assays performed in this study. Post-bronchodilator spirometry and gas transfer was performed according to ERS guidelines [20,21]. An 8 mL blood sample in a citrate vial was centrifuged to obtain approximately 5 mL of plasma which was stored at −80 °C pending analysis of Z-AAT polymers. In addition, some patients participated in a clinical study [22] for which serial blood samples were obtained for repeated measurements. Leftover plasma was also stored at −80 °C. We obtained cross sectional data from the Dutch part of the Alpha1 International Registry of patients who performed a pulmonary function test and donated a blood sample on the same day. We used the demographic data (Table 1) of 52 patients with ZZ phenotype, 5 with SZ phenotype and 6 with the homozygous rare AATD variant.

In addition, we included a cohort of 40 AATD patients with genotype ZZ enrolled in collaboration with the German Alpha1 Patient Association and Aachen University Alpha1-Liver Register. The institutional review board of Aachen University (EK 173/15) provided ethical approval. Demographic data are presented (Table 2).

### 2.2. Polymer ELISA

The AAT polymer ELISA using the monoclonal antibody LG96 (deposited under access No DSM ACC3092 at the German Collection of Microorganisms and Cell Cultures) was developed by Candor Bioscience GmbH, Wangen, Germany as previously described [7]. Nunc MaxiSorp flat-bottom 96-well plates (Thermofisher, Waltham, MA, USA) were coated overnight at 2–8 °C with monoclonal antibody LG96, at 1 µg/mL in coating buffer pH 7.4 (Candor Biosciences, Wangen, Germany). After a 2 h blocking step, the plasma samples were applied in the previously determined dilutions made in LowCross-Buffer (Candor Biosciences), which also served as a blank. Incubation was performed for 2 h at room temperature. For detection, the captured antigen was incubated with antibody (LG96)-horseradish peroxidase (HRP) conjugate (1:2000) for 2 h. The conjugate was prepared in advance with the HRP Conjugation Kit Lightning-Link (Abcam, Cambridge, UK) according to the manufacturer’s instructions. For signal development, SeramunBlau fast 2 microwell peroxidase substrate (Seramun, Heidesee, Germany) was used. The incubation was performed at room temperature for 12 min in the dark and the reaction was stopped with 2 M H_2_SO_4_. Plates were analyzed at 450 nm by microplate reader (Dynex Chantilly, VA, USA) equipped with Dynex Revelation 4.21 software. Measurements were carried out in triplicates and the lower limit of detection of the ELISA was 1 µg/mL.

### 2.3. Western Blot Analyses

Equal amounts of plasma were separated by 7.5% polyacrylamide gels under non-reducing conditions prior to transfer onto polyvinylidene difluoride membranes (Merck-Millipore, Burlington, MA, USA). Membranes were blocked for 1 h with 5% low fat milk (Carl Roth, Karlsruhe, Germany) followed by overnight incubation at 4 °C with specific primary antibodies: mouse monoclonal anti-AAT polymer antibody (clone 2C1, 1:500, Hycult Biotech, Uden, The Netherlands), or mouse monoclonal anti-AAT polymer antibody (LG96, 1:500). The immune complexes were visualized with anti-mouse HPR-conjugated secondary antibodies (DAKO A/S) and enhanced by Clarity Western ECL Substrate (BioRad, Hercules, CA, USA). Images were acquired using the Chemidoc Touch imaging system (BioRad) under optimal exposure conditions and processed using Image Lab v5.2.1 software (BioRad).

### 2.4. Statistical Analysis

Data are presented as mean ± SD, unless otherwise documented. Statistical analysis was performed in Graphpad Prism (version 8.4.2; San Diego, CA, USA) or Sigma Plot (version 14.5; San Jose, CA, USA). The Spearman (rs) or Pearson (r) correlation test was used for parameters and multiple regression analysis for analysis of interrelation between parameters.

## 3. Results

### 3.1. The Relationship between Plasma Z-Aat Polymers and Lung Function in Dutch Cohort

The FEV_1_ of the Dutch ZZ-AATD patients was 58 ± 23% pred and Kco was 61 ± 17% pred (Table 1). The correlation between both parameters was 0.57 (*p* < 0.0001). A weak inverse correlation was also found between patient’s age and FEV_1_ (r = −0.262) and gas transfer (r = −0.312), respectively. According to our ELISA results, plasma Z-AAT polymer levels were 947.5 µg/mL (median, 733.6–1218 µg/mL (95% CI)), however, there was no correlation between plasma Z-AAT polymer values and FEV1 % pred, or Kco % pred (Figure 1).

This pulmonary cohort was not designed to assess liver disease and so any associations must be exploratory. Nevertheless, we found no correlation between plasma polymers of Z-AAT and liver enzymes ALAT and GGT present in the medical records of these patients (Table 1). Ferrarotti et al. showed that levels of AAT were not affected by age, gender, or BMI [23]. In line with this, in this pulmonary cohort we found no correlation between Z-AAT polymer levels and patient age or gender (data not shown).

As a control, we measured plasma Z-AAT polymer levels in patients with phenotype SZ and genotype Mheerlen AATD variants (Figure 2).

Two patients with the homozygous Mheerlen variant had 1.9 and 2.0 µg/mL Z-AAT polymers, respectively. In the SZ group, the (n = 5) median Z-AAT polymer level was 250 µg/mL, with a range of 197–387 µg/mL. The levels of plasma Z-AAT polymers were below the lower detection limit in four patients with the null variants Q0bellingham, Q0soest, and Q0bredevoort, known for producing extremely low levels of AAT and in 10 patients with phenotype MM, a wildtype of AAT. Moreover, Respreeza, a plasma AAT product used for augmentation therapy (CSL Behring GmbH, Marburg, Germany) also had no Z-AAT polymer signal in our ELISA (data not shown).

So far, there are no studies establishing the temporal stability of circulating Z-AAT polymers. Hence, we determined Z-AAT polymer levels in repeated plasma samples from patient 1 (Pat 1) with normal FEV_1_ and Kco and patient 2 (Pat 2) with FEV_1_ 60% pred and Kco 45% pred. Notably, taking into account the wide range of Z-AAT polymer levels between ±100 ug/mL and ±1800 ug/mL (Figure 1), we considered that plasma levels of Z-AAT polymers were relatively stable when measured repeatedly for 14 consecutive days in these two ZZ-AATD patients (Figure 3).

Based on the fact that Z-AAT polymers are heterogenous and that different antibodies can recognize different profiles of polymers, we analyzed plasma samples from randomly selected ZZ AATD patients using 7.5% native PAGE following western blotting using anti-Z-polymer antibodies LG96 or 2C1 (Figure 4). Both antibodies are suitable for use in western blots for the Z-AAT polymer detection. However, one can clearly see that two antibodies recognize different polymer profiles in the same patient.

### 3.2. The Relationship between Plasma Z-AAT Polymers and Liver Function in the Alpha1-Liver Aachen Cohort

The characteristics of the cohort are presented in Table 2. In this cohort, plasma Z-AAT polymer levels were 1245.9 µg/mL (median, with 753–2034 µg/mL 95% CI). Out of all liver function measures, significant correlations were only found between plasma circulating Z-AAT polymers and GGT (rs = 0.57, *p* = 0.001), GLDH (rs = 0.4813, *p* = 0.0019), and TG (rs = 0.481, *p* = 0.0046) (Figure 5A–C). In this cohort, polymer levels directly correlated with patients age too (rs = 0.551, *p* = 0.0002) (Figure 5D). Remarkably, no correlation was found between ZZ AAT polymers and liver stiffness measures nor with BMI or the sex of patients.

Finally, we calculated the difference in plasma Z-AAT polymers between the lung and liver study population with the ZZ mutant of AATD. A Wilcoxon rank test showed a statistically significant difference (*p* < 0.0001), with higher Z-AAT polymer values for the liver group. This latter finding may be associated with the wider age range of the population of the liver cohort as compared to the lung cohort.

## 4. Discussion

The ZZ AATD individuals display an extraordinarily heterogeneous disease course. Some, especially smokers, are strongly predisposed to the development of lung emphysema during the fourth or fifth decade of life. The AATD-related liver diseases have a wide spectrum of presentations ranging from neonatal to adulthood fibrosis and end-stage cirrhosis requiring transplantation. The current clinical evidence suggests that both organ manifestations are largely independent and triggered by a combination of unique host and environmental factors [24]. This variability in clinical presentation prompted our further search for factors promoting/reflecting AATD-related diseases.

It is known that circulating polymers of AAT are present in patients with Z-AATD [10]; however, little is known about their role in the pathogenesis of AATD-related lung and liver diseases. For example, based on ATZ11 antibody ELISA, Aldonyte et al. showed significantly increased levels of Z-AAT polymers (2.4-fold, *p* < 0.001) in ZZ COPD patients compared to ZZ asymptomatic individuals [25]. However, authors did not report pulmonary function in this cohort. Tan et al. assessed levels of Z-AAT polymers in 244 ZZ subjects based on monoclonal 2C1 antibody-assay and found a significant association between Z-AAT polymers and FEV1/FVC [11]. By using the same 2C1 antibody-based assay, Núñez et al. reported a negative significant linear relationship between plasma Z-AAT polymer levels and airflow obstruction in a combined cohort of MZ and ZZ AATD patients [26]. Nevertheless, this latter study provided no data on the relationship between plasma AAT polymers and gas transfer values. Moreover, the same authors reported that patients with self-reported abnormal liver function had higher levels of Z-AAT polymers than those without a history of liver involvement and reported a correlation with liver stiffness as a non-invasive parameter of liver fibrosis. These findings led to the conclusion that levels of circulating Z-AAT polymers might be associated with the severity of lung and liver disease.

As a follow up to the above referred studies, we analyzed plasma levels of Z-AAT polymers in two independent retrospective cohorts of ZZ-AATD patients with lung and liver involvement, respectively, who did not receive AAT augmentation therapy. We measured plasma levels of Z-AAT polymers by using monoclonal LG96 antibody-based ELISA [7].

When interpreting our findings, it is very important to keep in mind that available mouse monoclonal antibodies against Z-AAT polymers, like LG96, 2C1 and ATZ11, recognize different conformational epitopes. For example, ATZ11 recognizes a conformational, non-linear epitope of Z-AAT and Z-AAT-elastase complex [10], whereas 2C1 is recognizes polymers formed by heating M- or Z-AAT at 60 °C. The 2C1 also recognizes polymers formed by the Siiyama (Ser53Phe) and Brescia (Gly225Arg) mutants and the His334Asp shutter domain mutant of AAT [17]. The LG96 antibody does not recognize polymers formed by M-AAT- or Z-AAT-elastase complexes and showed no cross reactivity with null variants of AAT such as Q0bellingham and Q0Soest [3]. Therefore, assays based on a different monoclonal antibody can give different values of circulating polymers.

Indeed, the range of plasma AAT polymers in ZZ AATD patients reported by Tan et al. [11] was 8.2–230.2 µg/mL (29- fold difference), and the range reported by Núñez [26] using the same 2C1 mAb was between 20–75 µg/mL, whereas in our assay plasma Z-AAT polymer levels had an 18-fold difference range between minimum and maximum value as shown in Figure 1. Hence, the quantification of circulating Z-AAT polymers remains an important issue in the field of AATD.

While we did not detect a significant correlation of polymer levels with lung function parameters, we observed a correlation with GGT and GLDH. GGT is an established marker of metabolic liver disease with an important role in defense against oxidative stress [27]. In ZZ individuals, elevated GGT is markedly associated with the presence of liver fibrosis [28]. GLDH is a mitochondrial enzyme that catalyzes the conversion of glutamate to 2-oxoglutarate [29] and reflects leakage from damaged or necrotic hepatocytes. Some studies suggest that GLDH activity is crucial for survival during hyperammonemia [30]. Notably, the relevance of elevated GLDH in ZZ individuals remains to be systematically studied. Overall, these data suggest that AAT polymer levels detected by the presented ELISA correlate with the extent of ongoing liver stress. The lack of correlation between polymer levels and LSM values should not be overemphasized as LSM only discriminates fairly between intermediate fibrosis stages and only four patients in our cohort had values indicative of advanced liver fibrosis. Finally, the presented analysis is impaired by the fact that LSM does not represent a linear function of liver fibrosis.

A direct relationship between TG and the levels of plasma Z-AAT polymers may potentially reflect mechanisms for the SERPINA1 Z allele-induced endoplasmic reticulum stress and autophagy dysregulation, which may aggravate proteotoxic stress and dysregulation in lipid metabolism from the accumulation of misfolded mutant AAT in the endoplasmic reticulum of hepatocytes [31]. It is known that concentrations of some liver enzymes, like ALP or GGT, as well as concentrations of liver-derived metabolites, such as bilirubin, may be influenced by metabolic processes beyond the liver [32]. We found no correlation between circulating Z-AAT polymers and ALP, ALT, AST, or bilirubin, suggesting that polymers might be particularly sensitive to metabolic liver stress or a stress in the periportal area of the liver.

Absence of a correlation between the Z-AAT plasma level and FEV_1_ or Kco lung function value may indicate that a blood sample taken at a random point in time of the course of lung disease in these patients may not reflect what has happened during the preceding period of lung destruction. Unfortunately, an insufficient number of patients had a known rate of decline of FEV_1_ to allow analysis of such rate with cross-sectional Z-AAT levels in plasma. Moreover, genotyping of AATD patients has not been a routine laboratory test in Leiden University Medical Center for many years. Therefore, our report is based on AAT phenotypes, and we cannot exclude that some patients with ZZ-AATD phenotype may have a different genotype. Unfortunately, many available plasma samples were too old to perform genotype analysis. Finally, patient serum CRP levels were not measured, and Z-AAT polymer levels may have been influenced by medications known to interfere with autophagy, e.g., metformin, statins, verapamil, resveratrol, and nortriptyline, but use of such medication was not specified in the database.

Based on our results, we speculate that reducing plasma levels of Z-AAT polymers in ZZ-AATD patients using the siRNA strategy may reduce liver damage, but it remains unclear whether or not this will also help to lower the progression of pulmonary emphysema [33]. In line with this, it was reported that siRNA treatment markedly reduces both serum and hepatic Z-AAT levels and in parallel, lowers serum ALT and GGT concentrations (LP10 (Abstract) Strnad P. et al. AASLD, The Liver Meeting Digital Experience. 2021).

## 5. Conclusions

Our data show that plasma levels of Z-AAT polymers are higher in ZZ-AATD patients with liver involvement rather than in those with lung involvement and suggest that high circulating levels of Z-AAT polymers may reflect liver but not the lung injury. Further studies are required to establish whether using circulating Z-AAT polymers as biomarkers is useful in predicting clinically important outcomes in individuals with ZZ AATD.

## Figures and Tables

**Figure 1 biomolecules-12-00380-f001:**
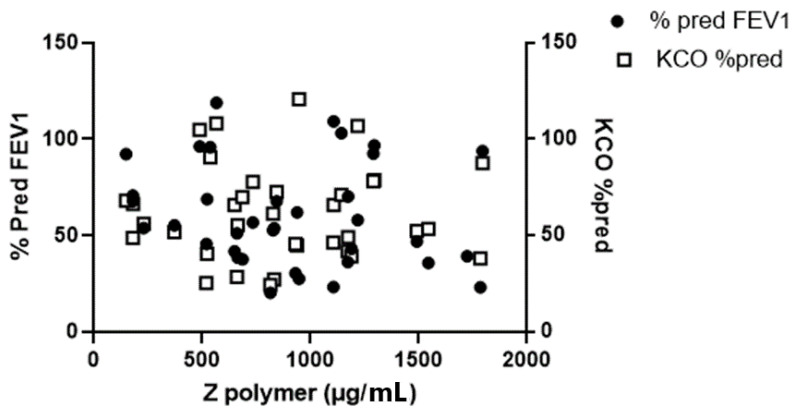
Pulmonary function of 52 ZZ-AATD patients was measured 15 min after the inhalation of 400 µg of salbutamol followed by spirometry and a CO diffusion gas transfer. Levels of plasma Z-AAT polymers (µg/mL) were measured by ELISA based on LG96 mAb (see materials and methods). There was no correlation between plasma Z-AAT polymer values and FEV1 % pred, or Kco % pred.

**Figure 2 biomolecules-12-00380-f002:**
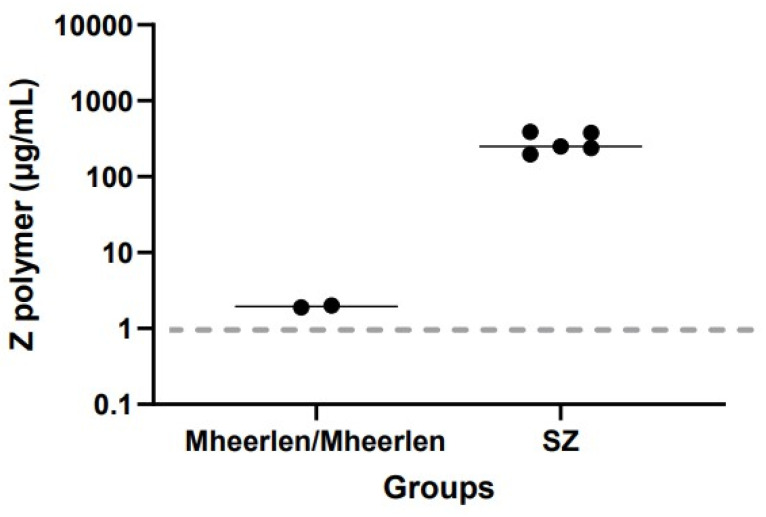
Z-AAT polymer levels in plasma of 2 patients homozygous Mheerlen genotype and 5 with phenotype SZ mutant of AATD. The dashed line represents the lower limit of detection of the assay. The line through the closed circles represents the median value.

**Figure 3 biomolecules-12-00380-f003:**
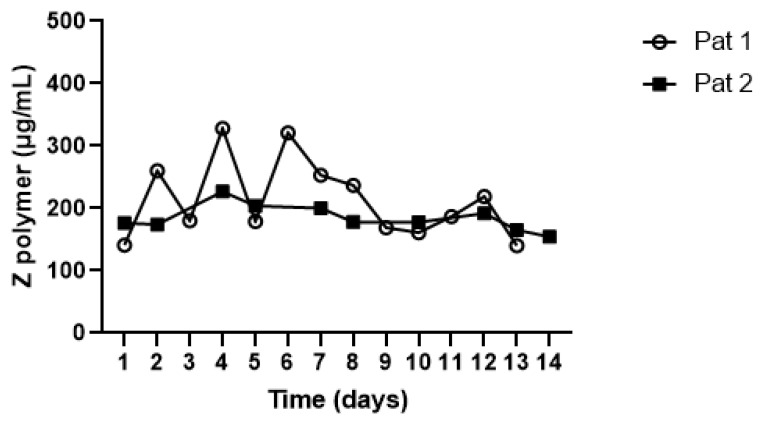
Repeatedly determined plasma levels of Z-AAT polymers in two randomly selected ZZ-AATD patients. On 14 consecutive days, plasma levels of Z-AAT polymers remained relatively stable in both patients.

**Figure 4 biomolecules-12-00380-f004:**
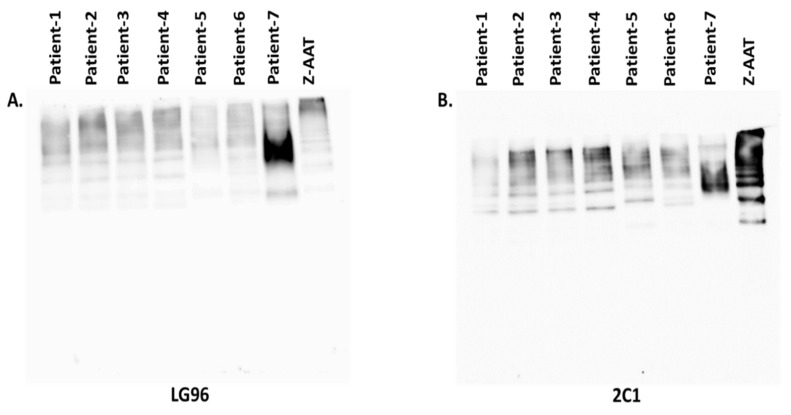
Z-AAT polymer analysis by Western blots using two different monoclonal antibodies. Equal amounts of plasma samples from seven ZZ-AATD individuals were electrophoretically separated using 7.5% native polyacrylamide gels followed by western blots. Purified polymeric Z-AAT isolated from pooled ZZ plasma samples was used as a positive control. The western blots were probed against mouse monoclonal anti-AAT polymer antibodies LG96 (**A**) and 2C1 (**B**)(both diluted 1:700). The image demonstrates that both antibodies recognize the Z-AAT polymers, but the profiles of recognized polymers differ. This blot is the representative from n = 2 independent repeats.

**Figure 5 biomolecules-12-00380-f005:**
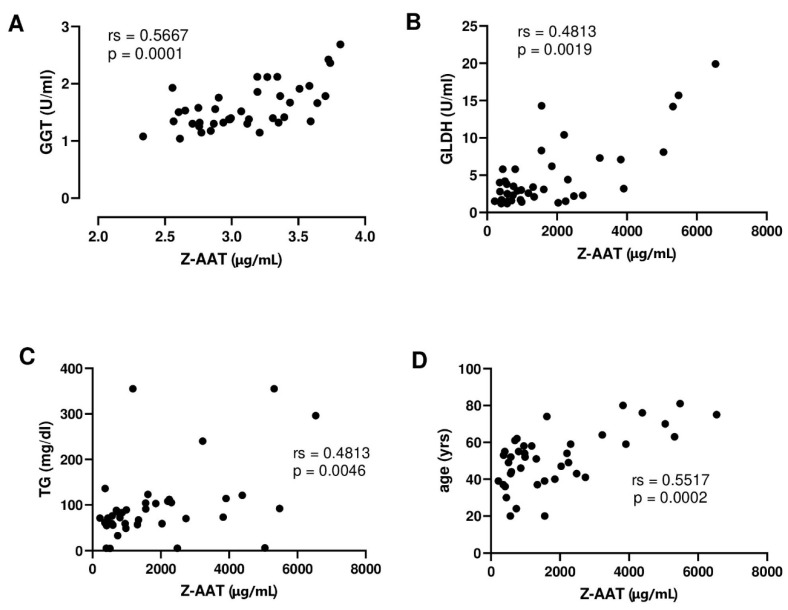
(**A**–**D**), Correlations between plasma levels of Z-AAT polymers and liver biomarkers and patient age: (**A**), the gamma-glutamyl transferase (GGT) values were log2 transformed; (**B**), glutamate dehydrogenase (GLDH), (**C**), triglycerides (TG), and (**D**) patient age Spearman correlation was calculated from 40 pairs of data and shows a significant correlation between plasma Z-AAT polymer levels and liver markers.

**Table 1 biomolecules-12-00380-t001:** Characteristics of the pulmonary patient cohort.

Characteristics	ZZ-AATD	SZ-AATD	Rare AATD
N male/female	26/30	2/3	2/4
Age (mean ± SD)	56 ± 7	51 ± 5	49 ± 9
FEV1 % predicted (mean ± SD)	58 ± 23	63 ± 17	53 ± 19
Kco % predicted (mean ± SD)	61 ± 27	60 ± 11	56 ± 18
Serum ALT > ULN (%)	0	0	0
Serum AST > ULN (%)	0	0	0
Serum GGT > ULN (%)	0	0	0

FEV1, forced expiratory volume in 1 s; Kco, constant for volume corrected gas transfer of carbon monoxide; ALT, alanine aminotransferase; AST, aspartate aminotransferase; GGT, gamma-glutamyl transferase; ULN, upper limit of normal. From 2 males and 2 females no ALT, AST, and GGT values were obtained and, therefore, these cases were excluded, leaving 52 patients for further analysis. Rare variants were Q0bellingham (n = 1), Q0soest (n = 1), Q0bredevoort (n = 2), and Mheerlen (n = 2).

**Table 2 biomolecules-12-00380-t002:** Characteristics of the liver patient cohort.

Characteristics	Pi*ZZ (n = 40), Mean ± SD
Age (years)	51.3 ± 15.3
Women (%)	15 (37.5)
BMI (kg/m^2^)	26.0 ± 4.2
BMI ≥ 30 (%)	7 (17.5)
Diabetes mellitus (%)	1 (2.5)

**Liver status**	.
Median stiffness (kPa)	7.8 ± 11.0
LSM ≥ 7.1 kPa (%)	7 (17.5)
LSM ≥ 10 kPa (%)	4 (10)
CAP (dB/m)	274.8 ± 50.8
ALT ≥ ULN (%)	9 (22.5)
AST ≥ ULN (%)	6 (15)
GGT ≥ ULN (%)	13 (32.5)
ALP ≥ ULN (%)	6 (15)
GLDH ≥ ULN (%)	9 (22.5)
Bilirubin (mg/dL)	0.65 ± 0.40
INR	1.07 ± 0.21
Triglycerides (mg/dL)	100.7 ± 63.8
HbA1c (%)	5.1 ± 0.3
Ferritin	185.1 ± 165.1
AAT serum level (mg/dL)	40.6 ± 30.4

BMI, body mass index; LSM, liver stiffness measurement; CAP, controlled attenuation parameter; ALT, alanine aminotransferase; AST, aspartate aminotransferase; GGT, gamma-glutamyl transferase; ALP, alkaline phosphatase; GLDH, glutamate dehydrogenase; INR, International Normalized Ratio or HbA1c, Hemoglobulin A1c; ULN, upper limit of normal of parameter. All categorical variables were presented as absolute (n) and relative (%) frequencies. Continuous variables were described as mean ± standard deviation (SD).

## Data Availability

Reasonable requests for data can be submitted to the corresponding author.

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
