# Peer review of "The Relationship between Plasma Alpha-1-Antitrypsin Polymers and Lung or Liver Function in ZZ Alpha-1-Antitrypsin-Deficient Patients"

_biomolecules, 2022, doi:10.3390/biom12030380_

Round 1

Reviewer 1 Report

It has been known for some years now that in the plasma of patients with PI*ZZ alpha1-antitrypsin (AAT) deficiency circulating polymers can be detected by means of a specific antibody. In this article, the authors try to understand whether levels of circulating Z-AAT polymers are associated with severity of lung and/or liver disease. The topic addressed is very interesting because of the little knowledge about the different aspects of the role of AAT polymers in the disease. However, the results are not so clear and leave some doubts about the argument.

The background is clearly written and turns out to be completed.

Materials and methods are well characterised but some points must to be cleared:

  1. In the “patient” section, “In addition, some patients participated in a clinical study [22] for which serial blood samples were obtained for repeated measurements”: do you have the informed consent for these subjects?;
  2. In the “patients” section, “patients with ZZ phenotype and SZ phenotype” means that genotype is not available? Genotype confirmation is suggested to avoid misdiagnosis. Moreover, specify here what Null variants carry the patients;
  3. In table 1 are reported 56 (26+30) ZZ patients while in the text and in figure 1 they are 52.

Results are quite confusing. In particular, in the control group are reported 4 Null subjects while in the materials and methods section and in table 1 six Null patients are considered; why 2 Null patients have been excluded in the results? Why the 6 SZ patients, previously cited in materials and methods and reported in table 1 are not analysed? Moreover, regarding the “10 patients with MM”, have they been genotyped to exclude the presence of other variants?

In figure 2, in the Y-axis (Z polymer), values are lower than those reported in the text; besides, polymers quantifications obtained for patients 1 seems to be not very stable, can you better explain this figure? Regarding last sentence of the results, the liver group doesn’t seem older than the lung cohort; can you better explain?

The authors, by figure 3, proved that antibodies LG96  and  2C1 recognize the Z-AAT polymers, but the profiles of recognized polymers are different. They also focused part of the discussion on this point “Therefore, assays based on different monoclonal antibody can give different values of circulating polymers”. Why the authors’ choice for polymers detection is LG96? With such theory it should be necessary to perform the experiments with both antibodies and compare the results. In fact, the range of polymers is highly wide in the 2 groups; are the data reliable? A very recent and similar paper (https://doi.org/10.1186/s12931-021-01842-5) ref26 reported completely different range of polymers detected by 2C1.

Discussion is generally well organised. At the end of the discussion the authors consider the siRNA strategy to reduce Z-AAT polymers; I suggest to expand this concept.

The english language and style are fine.

Other specific observations:

  • Line 25 check the brackets;
  • Line 26, check “polymers ranged between”, what is the range?;
  • Line 45 add Glu288Val after S mutation;
  • Line 73 “an” instead of “as”;
  • Line 134 “was” instead of “is”;
  • Line 159 check the brackets;
  • Line 209 check the brackets;
  • Line 225 “Z-AAT” is referred to polymers? specify;
  • Line 270 what “22” refers to?.

Author Response

Response to reviewer 1.

We are very grateful for the comments provided by the reviewer and for identifying some inconsistencies in the manuscript.

Materials and Methods.

Ad1. Participants in this study provided informed consent for the analysis of the samples for research purposes and the consent covers the analysis performed in this study

Ad2. Many samples from Leiden University Medical Center originated from more than 20 years ago. In those days genotyping of AATD patients was not a routine lab test in Leiden University Medical Center, while phenotyping was. Because we had phenotype results for all patients and AAT genotype of only part of the 56 patients, we decided to report based on AAT phenotype. We agree that some of these patients may have a different genotype. The specifics of the Null variants were provided in the manuscript on page 5, line 176. Two other Null variants were genotyped as homozygous Mheerlen. As this genotype is not based on a stopcodon, but an intracellular degradation of AAT protein, we did not report on those 2 Z-AAT values. Although we had planned to show all Z-AAT polymer levels from the 6 “Null” patients, limitation by the Journal of number of figures made us to decide not to show these. Based on the comments of the reviewer, we decided to add this figure and show the Z-AAT polymer values as closed circles of our Null and SZ patients. Most of these genotypes were detected in our samples by Prof Ferrarotti and therefore we acknowledged her at the end of the Conclusion of our manuscript.  All samples from Aachen were collected from 2015 onwards and were genotyped by Aachen.

Ad3. We are grateful for notifying us about the error of 56 patients in Table 1. The reason for this error is that originally the Leiden/Hannover authors had prepared a manuscript based only on Leiden patients and their association with pulmonary function. As an association was not detected we asked Dr Strnad to provide us with samples from ZZ-AATD patients that were collected in his study. As he agreed, we looked into LUMC clinical records for the availability of serum liver function tests (ALT, AST and GGT). From our 56 patients, those serum liver function tests were not available in 2 females and 2 males. We forgot to adjust the numbers in Table 1 accordingly and apologize for that. The legend to Figure 1 is adjusted in the revised manuscript.

Results.

We are grateful to the reviewer to be able to try to eliminate the confusion caused by the presentation of our results. We have added a Figure to show the Z-AAT polymer levels in the SZ and Null patients. The MM patients were phenotyped only and all MM samples were below the limit of detection of our ELISA. The Z-AAT polymer values are presented as individuals closed circles.

Results presented in Figure 2. The claim about “stability” of Z-AAT polymer levels of patient 1 is based on the wide range of Z-AAT polymer levels shown on the X-axis in Figure 1 which are between ± 100 ug/ml and ± 1800 ug/ml. Taking this wide range into consideration, we classified it as reasonably stable. We incorrectly used the word older population of liver cohort. When the age of liver and lung cohorts are compared, the age range of the liver population is much wider. Therefore, we adjusted the text on page 8, line 241 in the revised manuscript. The figure in the revised manuscript is now Figure 3.

Results of Figure 3. We have chosen the LG96 monoclonal antibody because it is recognizing an epitope that is not generated by heat-induced polymerization of the Z-AAT protein, does not recognize Z-AAT-elastase complexes and is based on an antibody sandwich approach. We wrote the paragraph starting at page 9, line 261 to explain the readers that absolute ELISA assay values generated by different antibodies very often show different values. However, what is important to interpret the data is to look at trends. The Z-AAT levels reported in the literature using different monoclonal antibodies show similar trends. This is also supported by Figure 4 of the revised manuscript. It was not the intention of our manuscript to show that our ELISA is better than the one based on the 2C1 monoclonal antibody. It is just different, but the difference in plasma values between MZ, ZZ and other deficient genotypes is still recognized by us, as stated between line 177 on page 5 to line 199 on page 6 of our revised manuscript.

Discussion.

We respect the suggestion by the reviewer to expand the concept of utilizing plasma Z-AAT polymer detection by ELISA when studying the effect of RNA silencing of the ZZ-AATD mutant. However, a manuscript by Dr Strnad of the results of an open label clinical trial using a compound for RNA silencing presented at the 2021 The Liver Meeting Digital Experience is in preparation and we wish not to elaborate on his findings as it may interfere with the reviewer process. Moreover, since the silencing approaches affect the total AAT production, it remains unclear whether assessing polymers would provide any benefits in comparison to assessing total AAT/Z-AAT levels.

Specific observations.

We have taken care of all items highlighted by the reviewer and the modifications are shown in the track changes modus in the revised manuscript.

Reviewer 2 Report

In the manuscript entitled “Relationship between plasma alpha-1-antitrypsin polymers and lung or liver function in ZZ alpha-1-antitrypsin deficient patients” the authors investigate the association between plasma AAT polymers and clinical variables related to Alpha-1 lung and liver disease. The AAT polymers level was measured by ELISA using a specific anti-Z polymers antibody, LG96, validated by previous work to differentiate between Z AAT polymers and normal M AAT or other mutated AAT molecules, including, S and Null variants.

The study addresses a relevant topic in the pathogenesis of Alpha-1 antitrypsin deficiency, the link between AAT polymers accumulation (gain-of-function) vs. lower AAT plasma levels (loss-of-function) and lung and liver tissue injury.

Based on the current study we learn that Z-AAT polymers are measurable in plasma of AATD patients with lung and liver related disease (distinct retrospective lung and liver cohorts). A new sandwich ELISA method is proposed, using a specific anti-Z polymers antibody, LG96 Cross-sectional Z-AAT polymers plasma level are associated with blood markers of liver disease, but not with pulmonary function test variables.

The manuscript could be improved by addressing several limitations in the approach and discussion

Major critiques:

  • Acute inflammatory state could significantly change AAT and Z-AAT circulating levels. Could the authors specify the status of patients included in the study, either by clinical measurements (time since last exacerbation) or laboratory measurements (CRP)?
  • Generation and clearance of Z-AAT polymers are linked to proper function of autophagy. Could the authors specify whether the patients included in the study were on medications known to interfere with the autophagy, e.g. metformin, statins, verapamil, resveratrol, nortriptyline?
  • The analysis and data interpretation would benefit from addition of a control group (ZZ or SZ AATD) that have no significant lung or liver disease
  • Could the authors discuss how cross-sectional Z-AAT polymers level might not be a biomarker of lung disease severity as measured by FEV1 or KCO but might perform better if tested as a biomarker of early lung disease or as a longitudinal biomarker of disease progression.
  • FEV1 and KCO correlate poorly with the severity and progression of Alpha-1 lung disease. Other clinical measurements, visual emphysema score, adjusted lung density, and rate of FEV1 decline/year have been proposed as better biomarkers of lung disease in Alpha-1 patients. Could the authors show if Z-AAT polymers are associated with rate of FEV1 decline or visual emphysema score.
  • Could the authors discuss further if the differences between Z-AAT polymers level in the liver vs. lung cohort could be related to the fact that the lung cohort included SZ and Null AATD individuals that present with lower / absent circulating polymers than the ZZ AATD patients.

Author Response

Response to reviewer 2.

We are very grateful for the comments provided by the reviewer and for suggestions to improve the manuscript.

Response to major critiques.

Ad1. We investigated our lung disease patients in our Outpatient Clinic when they were in a stable clinical condition and had no exacerbation of symptoms 4 weeks prior to the visit. This is now stated on page 2, line 83 in the revised manuscript. We did not measure CRP levels in serum.

Ad2. The co-medication mentioned by the reviewer and used by our patients was not specified in the clinical records for more than 40% of the pulmonary patients and for 25% of the liver patients.

Ad3. Patients with normal lung function were included in the analysis as shown in Figure 1. This Figure shows that there are 15 patients with FEV1 and Kco above 80% predicted.

Ad4. We added in the Discussion section on page 10, line 314 a paragraph stating: Absence of a correlation between the Z-AAT plasma level and FEV1 or Kco lung function value may indicate that a blood sample taken at a random point in time of the course of lung disease in these patients may not reflect what has happened during the preceding period of lung destruction.

Ad5. Unfortunately, we do not have visual emphysema scores or rate of decline of all study patients with a lung disease. This was added to the new paragraph as mentioned in our Ad4 response.

Ad6. The analysis of difference of Z-AAT plasma level between liver and lung patient cohorts was only performed in those studypatients classified by us as ZZ-AATD mutant. Therefore it is difficult for us the discuss further the influence by SZ and Null AATD as suggested by the reviewer.

Round 2

Reviewer 1 Report

“Ad1. Participants in this study provided informed consent for the analysis of the samples for research purposes and the consent covers the analysis performed in this study” You have to specify this in the text.

“Ad2. Many samples from Leiden University Medical Center originated from more than 20 years ago.” And you still have the informed consent after 20 years?

“In those days genotyping of AATD patients was not a routine lab test in Leiden University Medical Center, while phenotyping was. Because we had phenotype results for all patients and AAT genotype of only part of the 56 patients, we decided to report based on AAT phenotype. We agree that some of these patients may have a different genotype.” This is a limit of the study. If you have the blood of these patients you should perform genotype analysis.

“The specifics of the Null variants were provided in the manuscript on page 5, line 176. Two other Null variants were genotyped as homozygous Mheerlen. As this genotype is not based on a stopcodon, but an intracellular degradation of AAT protein, we did not report on those 2 Z-AAT values. Although we had planned to show all Z-AAT polymer levels from the 6 “Null” patients, limitation by the Journal of number of figures made us to decide not to show these.” If they are Mheerlen homozygous they are not Null..

“Based on the comments of the reviewer, we decided to add this figure and show the Z-AAT polymer values as closed circles of our Null and SZ patients.” In the new figure 2  on the X-axis you can’t write “0/0”, what does it means? Maybe better specify Mheerlen/Mheerlen (also in the figure 2 legend). Moreover, I think you have to add ZZ samples to this figure to have a comparison between the Z AAT polymer levels in the different groups.

“Ad3. The legend to Figure 1 is adjusted in the revised manuscript.” Line 110 put ALT instead of AST

“We are grateful to the reviewer to be able to try to eliminate the confusion caused by the presentation of our results. We have added a Figure to show the Z-AAT polymer levels in the SZ and Null patients. The MM patients were phenotyped only and all MM samples were below the limit of detection of our ELISA. The Z-AAT polymer values are presented as individuals closed circles.” The fact that the so-called MM patients have not been genotyped is a limit of the study. If you have the blood of these patients you should perform genotype analysis. Moreover, I don’t understand why do you mention here MM patients when they are not in the figure..

“Results presented in Figure 2. The claim about “stability” of Z-AAT polymer levels of patient 1 is based on the wide range of Z-AAT polymer levels shown on the X-axis in Figure 1 which are between ± 100 ug/ml and ± 1800 ug/ml. Taking this wide range into consideration, we classified it as reasonably stable.”  You have to specify this in the text.

“However, what is important to interpret the data is to look at trends. The Z-AAT levels reported in the literature using different monoclonal antibodies show similar trends.” If you should look at the trends and Z AAT polymer levels have wide ranges in ZZ patients, why do you report it as median?

Other specific observations:

3)            Line 45 add Glu288Val after S mutation: why do you specify “Z (Glu342Lys, also known as Glu366Lys)” and you don’t want to add “Glu288Val after S mutation”?

Reviewer 2 Report

In the revised manuscript entitled “Relationship between plasma alpha-1-antitrypsin polymers and lung or liver function in ZZ alpha-1-antitrypsin deficient patients” the authors have addressed all reviewer’s comments.

I would agree that this study tackles a relevant topic in the pathogenesis of Alpha-1 antitrypsin deficiency, the link between AAT polymers accumulation and lung and liver tissue injury using a novel ELISA method and ZZ and SZ Alpha-1 individuals with high level of circulating AAT polymers and adequate controls, Alpha-1 patients with mutations not associated with AAT polymers formation as well as healthy controls.

The revised manuscript would benefit before publication by inclusion of the following minor comments as study limitations:

  • That AAT polymers plasma level did not have concomitant CRP measurements.
  • That AAT polymer plasma level might be influenced by medications that were not available for the selected cohorts

Please correct that only ZZ Alpha-1 patients (line 258) were included in the comparison of plasma AAT polymer levels between the lung and liver cohorts.
